# Structure of Nanocrystalline, Partially Disordered MoS$_{2+\delta}$ Derived from HRTEM—An Abundant Material for Efficient HER Catalysis

**Emanuel Ronge** [1] , **Sonja Hildebrandt** [1], **Marie-Luise Grutza** [2], **Helmut Klein** [3], **Philipp Kurz** [2,*] **and Christian Jooss** [1,4,*]

1   Institute of Materials Physics, University of Göttingen, Friedrich-Hund-Platz 1, 37077 Göttingen, Germany; emanuel.ronge@phys.uni-goettingen.de (E.R.); s.hildebrandt01@stud.uni-goettingen.de (S.H.)
2   Institute for Inorganic and Analytical Chemistry and Freiburg Material Research Center (FMF), University of Freiburg, Albertstraße 21, 79104 Freiburg, Germany; marie-luise.grutza@ac.uni-freiburg.de
3   GZG Crystallography University of Göttingen, Goldschmidtstr. 1, 37077 Göttingen, Germany; hklein@uni-goettingen.de
4   International Center for Advanced Studies of Energy Conversion (ICASEC), University of Göttingen, D-37077 Göttingen, Germany
*   Correspondence: philipp.kurz@ac.uni-freiburg.de (P.K.); cjooss@gwdg.de (C.J.); Tel.: +49-0761-203-6127 (P.K.); +49-551-39-25303 (C.J.)

**Abstract:** Molybdenum sulfides (MoS$_x$, x > 2) are promising catalysts for the hydrogen evolution reaction (HER) that show high hydrogen evolution rates and potentially represent an abundant alternative to platinum. However, a complete understanding of the structure of the most active variants is still lacking. Nanocrystalline MoS$_{2+\delta}$ was prepared by a solvothermal method and immobilized on graphene. The obtained electrodes exhibit stable HER current densities of 3 mA cm$^{-2}$ at an overpotential of ~200 mV for at least 7 h. A structural analysis of the material by high-resolution transmission electron microscopy (HRTEM) show partially disordered nanocrystals of a size between 5–10 nm. Both X-ray and electron diffraction reveal large fluctuations in lattice spacing, where the average c-axis stacking is increased and the in-plane lattice parameter is locally reduced in comparison to the layered structure of crystalline MoS$_2$. A three-dimensional structural model of MoS$_{2+\delta}$ could be derived from the experiments, in which [Mo$_2$S$_{12}$]$^{2-}$ and [Mo$_3$S$_{13}$]$^{2-}$ clusters as well as disclinations represent the typical defects in the ideal MoS$_2$ structure. It is suggested that the partially disordered nanostructure leads to a high density of coordinatively modified Mo sites with lower Mo–Mo distances representing the active sites for HER catalysis, and, that these structural features are more important than the S:Mo ratio for the activity.

**Keywords:** hydrogen evolution catalysis; molybdenum sulfides; nanocrystalline materials

## 1. Introduction

The production of hydrogen from renewable energy sources by the splitting of water is a clean alternative to fossil fuels [1–3]. For a large scale application of water-splitting electrolyzers, abundant, stable and efficient electrocatalysts for the hydrogen evolution reaction (HER) are needed. However, the best known materials for this purpose in acidic conditions are noble metals, such as for example, platinum. Because of the low abundance of noble metals and their resulting high price, the search for more affordable alternatives is of high interest [4,5].

Due to their relatively low overpotentials and the good availability of Mo, molybdenum sulfides have gained increasing attention in HER catalyst research during the past few decades [5]. However,

it is by now accepted that the catalytic activity of crystalline $MoS_2$ in its most common 2H modification is limited to about 5 mA cm$^{-2}$ at an overpotential $\eta \sim 500$ mV [6] because of the special nature of the HER active sites, which are mainly present at the edges of the two-dimensional $MoS_2$ planes [7–9]. In contrast, for the 1T polymorph which has a higher HER activity, the crystalline basal planes might be active sites [9,10]. However, this system is not stable.

In order to improve catalytic activity, sulfur-rich "$MoS_x$" materials (with x > 2) have been synthesized and this approach has yielded very promising results [4,11,12]. Some $MoS_x$ electrodes can deliver current densities of 10 mA cm$^{-2}$ at $\eta \sim 170$ mV in 0.5 м $H_2SO_4$. In special applications, for example, acidic industrial wastewaters, these catalysts also show a much better long-term stability than Pt [13]. $MoS_x$ materials can be synthesized by solvothermal synthesis [12,14], wet chemical synthesis [5,15], electrodeposition [4,16], thermal decomposition [17] or chemical oxidation [16]. Typical precursors include ammonium tetrathiomolybdate (($NH_4)_2[MoS_4]$) as a solid [12,14,16], aqueous solutions containing $[MoS_4]^{2-}$ [4,16] or ammonium heptamolybdate (($NH_4)_6[Mo_7O_{24}]$) [5] or $MoO_3$ reacting with $NaS_2$ [15].

These different synthesis routes for $MoS_x$ generally lead to stoichiometries with 2 < x < 4 and highly disordered, X-ray amorphous structures [4,11,14,18]. X-ray photoemission spectroscopy (XPS) indicates the presence of a variety of different sulfur species in the form of unsaturated, terminal, bridging and maybe even apical sulfides and disulfides [4,11,12,14,15].

As results from these studies, two quite different structure models for $MoS_x$ have been proposed. One model suggests a polymer like structure with chains of $[Mo_2S_9]$- [11,18–21] or $[Mo_3S_{13}]$-units [16,18,22]. Other models feature a disordered arrangement of $[Mo_3S_{13}]$-clusters [11,12,18,19,23,24]. Sometimes, crystallization of the amorphous phase to $MoS_2$ nanoparticles during HER is reported [5,25], indicating a structural affinity to $MoS_2$.

The polymer model is supported by a publication of Tran et al. [16] using real space scanning transmission electron microscopy (STEM) imaging of a $MoS_x$ sample with a relatively large S:Mo ratio of about 4. However, the structural analyses of other $MoS_x$ with lower sulfur contents (2 <x < 4) were mostly based on X-ray diffraction (XRD) which does not provide precise structural information for partially disordered systems. Thus, precise high-resolution real space information about $MoS_x$ structures is of crucial importance for a better understanding of the HER activity and the identification of active site(s).

The nature of the active sites for disordered $MoS_x$ with x > 2 ($MoS_{2+\delta}$) is controversially discussed in the literature. Depending on the suggested structural model, different hypotheses exist. If a $MoS_2$-like nanostructure arises during HER, there is evidence that terminal disulfides ($S_2^{2-}$) at the edges of the nanocrystals act as active centers [5,25]. However, no crystalline edges are present in amorphous $MoS_{2+\delta}$ but here coordinatively unsaturated molybdenum or sulfur sites are considered to act as HER active sites [4]. Moreover, density functional theory indicates a higher activity of bridging $S_2^{2-}$ which would explain the excellent performance of $[Mo_3S_{13}]^{2-}$ clusters [11,26]. That $S_2^{2-}$ may be an active site for $MoS_{2+\delta}$ is also supported by XPS studies [12]. Consequently, a better understanding of the actual HER mechanism of disordered $MoS_x$ requires a comprehensive structure model.

Herein, we report on a detailed structural analysis of two solvothermally synthesized $MoS_{2+\delta}$ samples and their hydrogen evolution activity compared to $MoS_2$, using different electrochemical measurements (CV, CP and Tafel analyses), high-resolution transmission electron microscopy (HRTEM), as well as electron- and X-ray diffraction. HRTEM shows that the "X-ray amorphous" structure is in fact nanocrystalline and features a pronounced disorder within the individual nanocrystals. The detected fluctuations of in-plane and out of plane lattice parameters measured by X-ray diffraction (XRD), electron diffraction and HRTEM agree very well. These observations serve as the basis for the development of a three-dimensional structural model for $MoS_{2+\delta}$, which is qualitatively consistent with spectroscopic and structural information about $MoS_x$ for 2 < x < 4 from literature.

## 2. Results

### 2.1. Electrochemical Characterization of MoS$_{2+\delta}$ Electrodes

Electrodes were prepared by immobilizing two synthetic MoS$_x$ (with x = 2.6 and 3.4, respectively) and commercially available 2H-MoS$_2$ on graphene. Cyclic voltammograms (CVs), Tafel plots and chronopotentiometric measurements (CPs) in a strongly acidic electrolyte (0.5 м sulfuric acid, pH 0.3) were used as descriptors for the electrocatalytic performance (Figure 1). In cyclic voltammograms, the onset potentials for HER at 3 mA cm$^{-2}$ were with a value of −190 mV similar for both synthetic MoS$_x$ materials, while MoS$_2$ showed a significantly higher onset potential (−365 mV) and therefore lower catalytic activity.

Another common descriptor for the HER activity is the overpotential ($\eta$) needed to reach current densities of 10 mA cm$^{-2}$ in the CVs (Table 1). MoS$_{2.6}$ showed the best performance with an overpotential of $\eta$ = 235 mV, followed by MoS$_{3.4}$ with $\eta$ = 245 mV and MoS$_2$ with $\eta$ = 450 mV. This is in agreement with the literature [4,6], where it is also described that disordered MoS$_x$ are far more active HER catalysts than crystalline MoS$_2$. For all of these values, one has to keep in mind that the current densities are derived for the geometric area of the electrodes. However, the difference in activity between the prepared samples could also lie in their different stoichiometric compositions. Assuming a molybdenum-based proton reduction mechanism as postulated for example, by Tran et al. [16], defective structures and molybdenum-rich materials would be in favor of high proton reduction activity. Indeed, when the detected currents are normalized to the amount of molybdenum on the electrodes, MoS$_{2.6}$ and MoS$_{3.4}$ show nearly identical overpotentials of $\eta$ = 250 and 255 mV at 10 mA μmol(Mo)$^{-1}$, while MoS$_2$ is not able to reach this current (see Figure S7). Concluding from these results, a high sulfur content seems to be much less important for the HER catalysis rate than the existence of a disordered structure for 2 < x < 3.5.

In addition, the particle sizes of the synthesized materials differ with (0.18 ± 0.06) μm significantly from the commercial MoS$_2$ with particle sizes of (0.8 ± 0.9) μm showing a much broader dispersity of the particles (see Figure S6). Hence, the prepared MoS$_x$ both contain more accessible active sites than MoS$_2$ where the active sites are believed to be the structural sulfur-rich edges [7]. However, the different particle sizes and shapes result in an increase of the MoS$_{2+\delta}$ surface by a factor of only about 1.9 compared to MoS$_2$ (see ESI for details). Assuming the same degree of porosity, this indicates that the higher activity of the MoS$_{2+\delta}$ electrodes compared to the MoS$_2$ electrode cannot be explained by morphology effects alone.

Tafel slopes were calculated from chronoamperometric "staircase measurements" (see materials and methods) and values of 100 mV dec$^{-1}$ for MoS$_{2.6}$, 90 mV dec$^{-1}$ for MoS$_{3.4}$ and 160 mV dec$^{-1}$ for MoS$_2$ were obtained for the same catalyst loadings. According to these values, the rate limiting step for MoS$_2$ might be the adsorption of an H atom (the Volmer reaction), while the results for MoS$_{2.6}$ and MoS$_{3.4}$ do not give clear evidence on the rate limiting step as the values lie between the boundary values of 40 mV dec$^{-1}$ for the Heyrovsky reaction (reductive desorption) and the Volmer reaction (>120 mV dec$^{-1}$) [13].

All of the measured values for overpotentials and Tafel slopes of the MoS$_x$ studied here cannot compete with the best MoS$_x$ that are known today (Table 1), which is hardly surprising given the fact that both material synthesis and electrode fabrication were not optimized. However—and most important for the following detailed structural characterization—both MoS$_x$ samples show very respectable electrocatalytic HER activity while the MoS$_2$ reference does not.

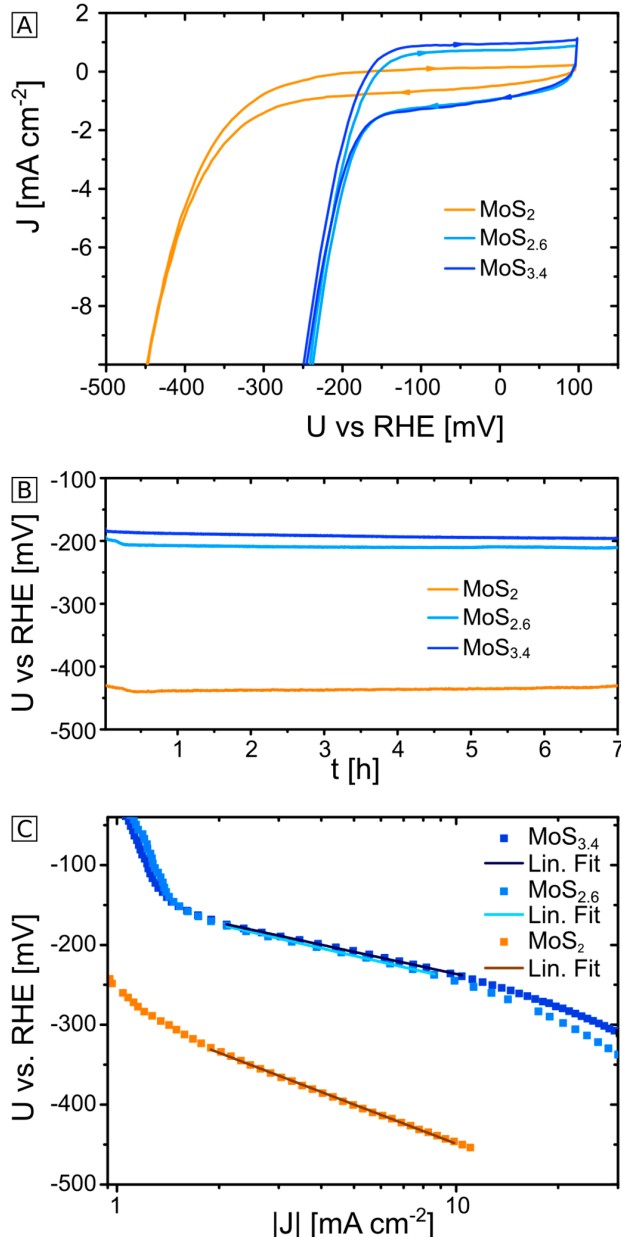

**Figure 1.** Electrochemical measurements for the two MoS$_x$ and MoS$_2$ (immobilized on graphite) in sulfuric acid (0.5 м, pH 0.3). (**A**): cyclic voltammetry (cycle 6, scan rate 20 mV s$^{-1}$); (**B**): chronopotentiometry at a current density of 3 mA cm$^{-2}$, the first 2 min are cut off; (**C**): Tafel analysis. See materials and methods for experimental details.

The long-term stability of the electrodes was tested in CP measurements (Figure 1B). Herein, both synthetic MoS$_x$ materials clearly outperformed crystalline molybdenum disulfide over a period of 7 h. For a set current density of 3 mA cm$^{-2}$, at the end of the experiments MoS$_{2.6}$ and MoS$_{3.4}$ showed overpotentials of ~210 mV and ~195 mV, respectively, in comparison to ~430 mV for MoS$_2$. In these measurements, both synthetic MoS$_x$ showed extremely stable performances with negligible increases of the overpotentials. The difference in the catalytic activity of MoS$_x$ might be due to some MoS$_2$ particles as the powder diffractogram shows a sharp peak at (0 0 2) for MoS$_{2.6}$. Over time, MoS$_2$ showed a slight increase of the HER overpotential, which might be due to a reduced electric resistivity of the catalyst layer related to a shrinking thickness and/or structural rearrangements of the surface leading to a higher amount of active sights.

**Table 1.** Tafel slopes and overpotentials at 10 mA cm$^{-2}$ of MoS$_x$ and MoS$_2$.

|  | Tafel Slope [mV dec$^{-1}$] | $\eta$ @ 10 mA cm$^{-2}$ [mV] | $\eta$ @ 10 mA µmol(Mo)$^{-1}$ [mV] |
|---|---|---|---|
| **MoS$_{2.6}$** | 100 | 235 | 250 |
| **MoS$_{3.4}$** | 90 | 245 | 255 |
| **MoS$_2$** | 160 | 450 | - |
| **MoS$_2$ [27]** | 160 | - | - |
| **MoS$_2$ (step-edged stacks) [28]** | 59 | 104 | - |
| **1T MoS$_2$ (porous) [29]** | 43 | 153 | - |
| **MoS$_{2+x}$ [4]** | 40 | 160 | - |
| **MoS$_{3.5}$ [5]** | ≈60 | ≈200 | - |

### 2.2. Electron and X-ray Diffraction Analysis of the MoS$_{2+\delta}$ Structure

For the determination of the structure of MoS$_{2+\delta}$, XRD measurements were carried out. The powder diffractograms for the two MoS$_{2+\delta}$ samples (MoS$_{2.6}$ & MoS$_{3.4}$) from different batches are given in Figure 2. For comparison, a powder XRD of the crystalline MoS$_2$ sample was measured as well. A Rietveld refinement of the MoS$_2$ data was conducted, which yielded results in good agreement with the structural model of Wildervanck et al. [30]. The results are shown in the ESI in Figure S1 and Tables S2 and S3. In contrast to MoS$_2$, the XRD reflexes of MoS$_{2+\delta}$ are strongly broadened. Both samples show a very wide peak at about 8 Å which deviates about 1–2 Å from the literature value for MoS$_2$. Besides this difference, the centers of the other reflexes of the MoS$_{2+\delta}$ XRD pattern fit the positions of the MoS$_2$ data (see Table S1). The main difference among the MoS$_{2+\delta}$ samples themselves is the sharp double peak of sample MoS$_{2.6}$, which reflects the presence of some larger more MoS$_2$-like grains. This might indicate a convergence of the MoS$_{2+\delta}$ structure towards MoS$_2$ for decreasing S:Mo ratios. The presence of only one sharp MoS$_2$ reflex (002) in MoS$_{2.6}$ can be explained by a preferred orientation of MoS$_2$ –like crystals due to their plate like morphology (see Figure S6A).

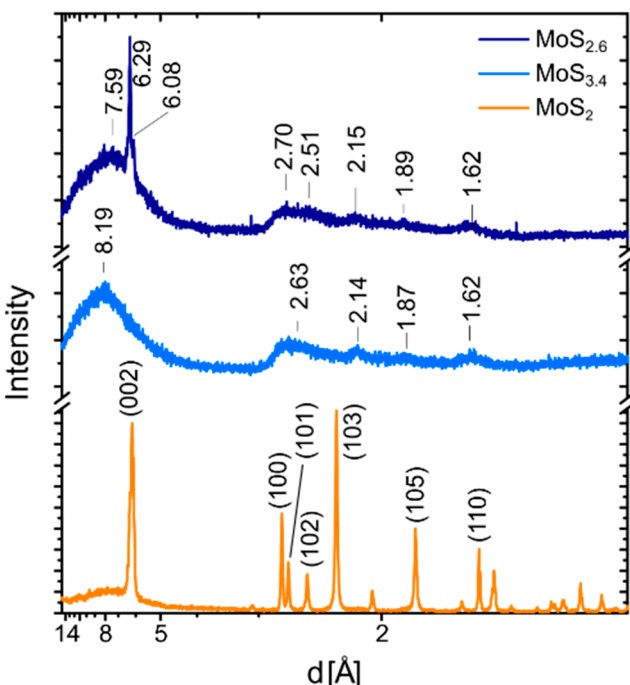

**Figure 2.** Powder X-ray diffraction (XRD) pattern of MoS$_2$ and the nanocrystalline, partially disordered samples MoS$_{2.6}$ and MoS$_{3.4}$. The indicated diffraction maxima are determined by fitting of a gaussian function.

In addition to the XRD analysis, selected area electron diffraction (SAD) was carried out. The electron diffraction patterns of the two samples $MoS_{2.6}$ and $MoS_{3.4}$ are shown in Figure S2 and consist of only of rings, while no spots are visible. By circular integration of the intensities, the profiles shown in Figure 3 were obtained. Three very broad reflexes are visible in the intensity profiles of both samples while the values for $MoS_{2.6}$ are shifted to slightly lower values. Nevertheless, both are consistent with the $MoS_2$ structure within the measurement accuracy. A detailed discussion of the peak positions of $MoS_{2+\delta}$ compared to $MoS_2$ [30,31] can be found in the supplement (see Table S4).

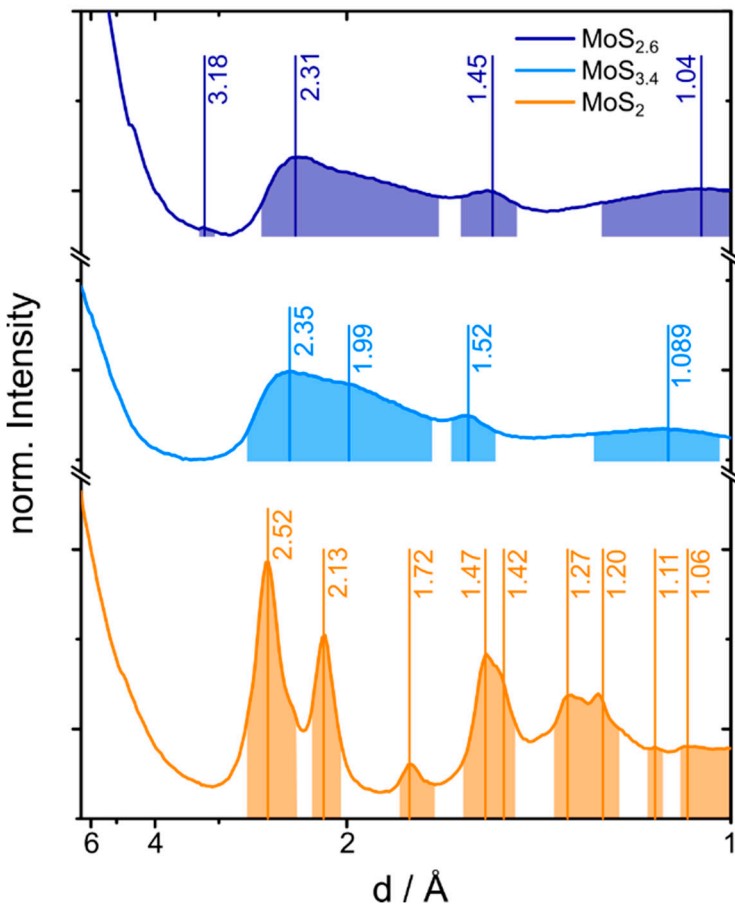

**Figure 3.** Intensity profiles of the representative electron diffraction patterns of the two $MoS_{2+\delta}$ batches investigated in this study ($MoS_{2.6}$ in dark blue & $MoS_{3.4}$ in light blue). Vertical lines indicate the positions of diffraction maxima and the marked areas indicate the width of the reflexes. Note the reflex at $\approx 4.53$ Å is most likely an artefact due to the beam stopper.

The results of the X-ray and electron diffraction experiments indicate that the structure of the $MoS_{2+\delta}$ samples is comparable and compatible with a disordered $MoS_2$ structure independent of their S:Mo ratio. The shift of the first peak of the $MoS_{2+\delta}$ samples compared to the (002) reflex of $MoS_2$ towards lower angles and the variation of the peak center reflects an enlargement and fluctuations of the lattice parameter in [001] direction.

*2.3. HRTEM Analysis of the Microstructure of $MoS_{2+\delta}$*

To gain a deeper insight into the structure, HRTEM analysis was performed. During the HRTEM analysis, no significant changes to the crystal structure was observed over time at a beam dose rate of 10,000 to 64,700 $e^-/(Å^2 s)$. Thus, we find that $MoS_{2+\delta}$ is stable under the electron beam which is a requirement for a reliable HRTEM analysis. The HRTEM images of $MoS_{2+\delta}$ reveal a bended and partially disordered crystal layer stacking. Coherence length of ordered stacking along the (001)

direction is about 5 nm and ordering within the planes is reaching up to 10 nm. The in-plane ordering can also be observed along [001] zone axis: A HRTEM image of such a crystal plate is presented in Figure 4.

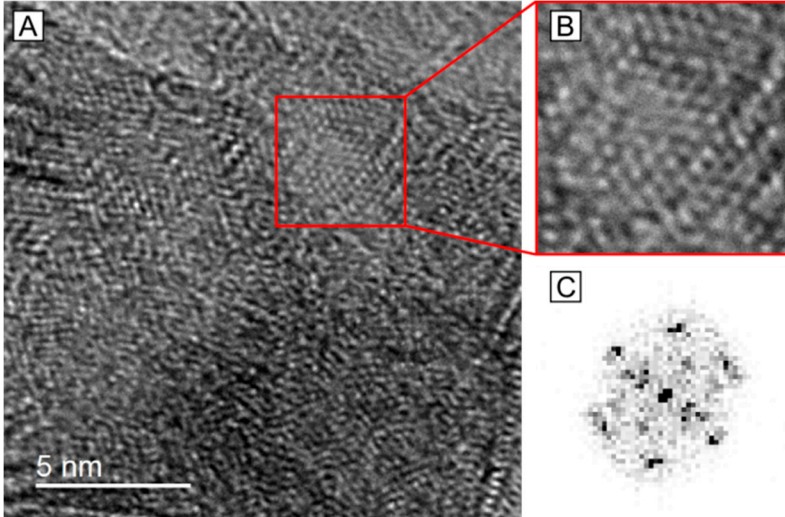

**Figure 4.** High-resolution transmission electron microscopy (HRTEM) image of sample $MoS_{3.4}$ showing the in-plane ordering. (**A**): Overview of the edge of a particle; (**B**): zoom in the area marked with a red square in A; (**C**): Fast Fourier Transformation (FFT) of the area shown in B.

Fast Fourier Transformation (FFT) shows the typical hexagonal symmetry of the (100) lattice planes of $MoS_2$. The discovery of this ordering with a hexagonal lattice symmetry of atomic positions within the layers as well as an ordering of the layers along the (001) direction support our conclusion that $MoS_{2+\delta}$ and $MoS_2$ show a close structural affinity. The correlation length along (001) is $\xi_c \approx 5$ nm, while from the in-plane ordering an in-plane correlation length of $\xi_{ab} \approx 4$ nm can be determined. The reason for the deviation from the prior determined length of $MoS_{2+\delta}$ planes of 10 nm is the bending of the crystals planes which disturbs the phase contrast in the top down view. Such a correlation length in the order of 1–5 nm is also called medium range order and can be found in nanocrystalline [32] as well as in amorphous systems [33]. From the XRD measurements, a correlation length of $(1.22 \pm 0.10)$ nm can be calculated with the Scherrer-Formula. It is well known that the Scherrer-Formula underestimates the grain size for partially disordered nanoparticles, since it measures the coherence length of the lattice [34].

A more detailed analysis of the HRTEM images for $MoS_{2+\delta}$ was carried out using FFT and its results are exemplified in Figure 5 C. The lattice distance $d_{(100)}$ exhibits a general trend of being locally reduced compared to the literature values of $MoS_2$ ($d_{\{100\}} = 2.7368$ Å [30]) with an average value of $d_{(100)} = (2.69 \pm 0.21)$ Å and local variations from 2.45 Å to 2.79 Å (see also Table S5 in the ESI for a summary). Within the measurement accuracy and statistics no significant difference between the two S:Mo ratios can be observed. Both samples ($MoS_{2.6}$ & $MoS_{3.4}$) show similar reduction and local variations in the Mo–Mo distance.

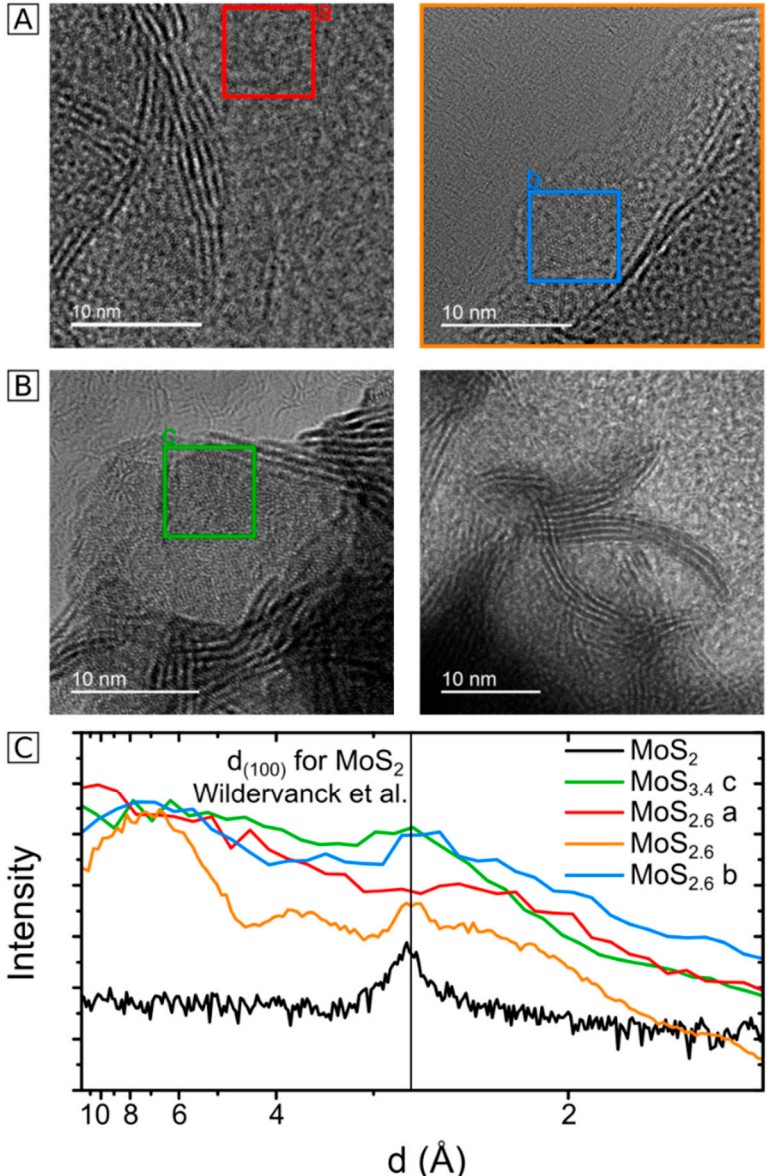

**Figure 5.** HRTEM analysis of nanocrystalline $MoS_{2+\delta}$. (**A**): HRTEM images of $MoS_{2.6}$; (**B**): HRTEM images of $MoS_{3.4}$; (**C**): representative FFT intensity profiles. The FFT of $MoS_2$ is taken from Figure S4A, and the other underlying FFTs can be found in Figure S5; The in-plane lattice spacing of $MoS_2$ $d_{100} = 2.737$ Å (Wildervanck et al. [30]) is indicated. The region of interest of the FFT plots is marked with a rectangular in subfigure A and B with the corresponding color.

A close up of a cross section image of the crystal layers is shown in Figure 6A. The layers are not perfectly flat and parallel which leads to a variation in layer distances. On average, the lattice stacking distance $d_{(001)}$ of $MoS_{2+\delta}$ varies from 11.15 Å to 15.93 Å. The overview in Table S5 indicates that $d_{(001)}$ is generally enlarged when compared to the literature value of $MoS_2$ ($d_{\{001\}} = 12.294$ Å) [30] for both S:Mo ratios, which is in good agreement with XRD analysis. Next to dislocations also disclinations are visible in Figure 6B (see also Figure S4B). This is quite unusual for solid state materials but, for example, has been observed in fullerene-like variants of $MoS_2$ by Srolovitz et al. [35].

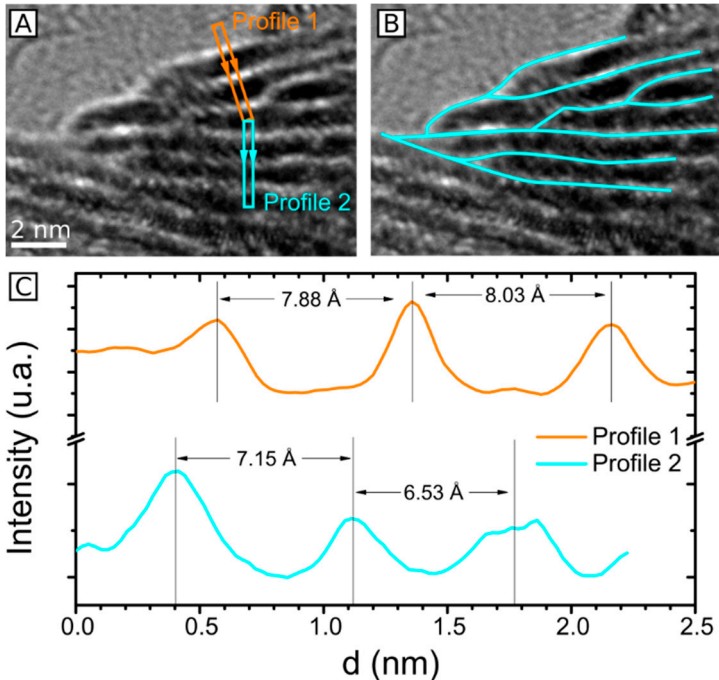

**Figure 6.** Close up of the HRTEM image from the left side of Figure 5 B, showing the $MoS_{3.4}$ lattice planes in cross section view. (**A**): Position of the intensity profile with a width of 10 px and direction marked by arrows. (**B**): Disclinations formed by merging $MoS_x$ layers are highlighted by cyan lines. (**C**): Intensity profiles indicated in A and exemplary variations in lattice distance. For the full analysis see Table 2 and Table S5.

In order to provide accurate calibration of the HRTEM images, all images were calibrated by measurements of gold particles. In addition, the crystalline $MoS_2$ sample was also analyzed by HRTEM, as shown in Figure S4. Both average lattice parameters $d_{(100)}$ and $d_{(001)}$ are in agreement to the literature values [30] within the accuracy (see experimental) as well as to our XRD data.

In some cases, a recrystallization of amorphous $MoS_x$ under the electron beam was observed [36,37]. In addition, Xi et al. [37] report on the transformation of amorphous $MoS_x$ after 2 h of HER to a nanocrystalline material which is stable under the electron beam. The crystal diameter is comparable to the ones shown in Figures 4 and 5. However, they observe a lower degree of stacking of crystal planes compared to Figure 6, indicating a higher degree of order in our system. The transformation of the amorphous to nanocrystalline $MoS_x$ in Xi et al. [37] was also accompanied by an increase in hydrogen production indicating that the nanocrystalline phase is the more active and stable configuration. Together with our results this suggests that our two $MoS_{2+\delta}$ samples both represent a thermodynamic stable and active form of $MoS_x$.

## 3. Structure Model

The combined application of electron and X-ray probes reveal that $MoS_{2+\delta}$ exhibits a nanocrystalline, partially disordered $MoS_2$-like structure with a locally reduced lattice parameter *a* and an enlarged mean parameter *c*, with both showing strong fluctuations. This follows from the observed hexagonal in-plane symmetry of the crystallites, the *c*-axis stacking of planes and the measured fluctuations of the lattice distances revealed by HRTEM as well as diffraction techniques. A suggested structural model for $MoS_{2+\delta}$ consistent with these results is depicted in Figure 7. Table 2 compares the measured mean lattice spacing in [001] and [100] directions as well as their fluctuations with the literature values for $MoS_2$. The results for $d_{100}$ from TEM are in good agreement with the X-ray diffraction analysis. In addition to the small grain sizes of a few nm, the strong disorder within the nanocrystals is consistent with the broad reflexes in XRD and SAD.

**Table 2.** Comparison of the (100) and (001) lattice distances of $MoS_{2+\delta}$ found by transmission electron microscopy (TEM) and XRD with literature values for $MoS_2$ [30]. The XRD uncertainty is determined by the full width at half maximum (FWHM).

| | Literature [30] | HRTEM $MoS_{2+\delta}$ | | | XRD |
|---|---|---|---|---|---|
| | $MoS_2$ | Mean | Min | Max | |
| $d_{(100)}$ [Å] | 2.7368 | 2.69 ± 0.21 | 2.45 ± 0.18 | 2.79 ± 0.03 | $2.69^{+0.16}_{-0.39}$ |
| $d_{(001)}$ [Å] | 12.294 | 12.9 ± 0.8 | 11.15 ± 0.19 | 15.93 ± 0.61 | $15^{+10}_{-4}$ |

In agreement with the stoichiometry of $MoS_{2+\delta}$, the excess sulfur must be present in the form of unsaturated, terminal, bridging and possibly apical sulfides and disulfides, as visible by XPS [4,11,12,14,15]. This modification results in structural changes relative to the 2H-$MoS_2$ structure. Our model suggests that some parts of the structure show similarities to the atomic arrangements found in $[Mo_2S_{12}]^{2-}$ and $[Mo_3S_{13}]^{2-}$ clusters. Like $MoS_{2+\delta}$ these clusters have also a reduced Mo–Mo distance compared to $MoS_2$ (see Table 3). The disorder in the in-plane lattice distance of $MoS_{2+\delta}$ thus reflects local sulfur-rich disorder in the form of cluster like structural units which are incorporated into the $MoS_2$ nanocrystals.

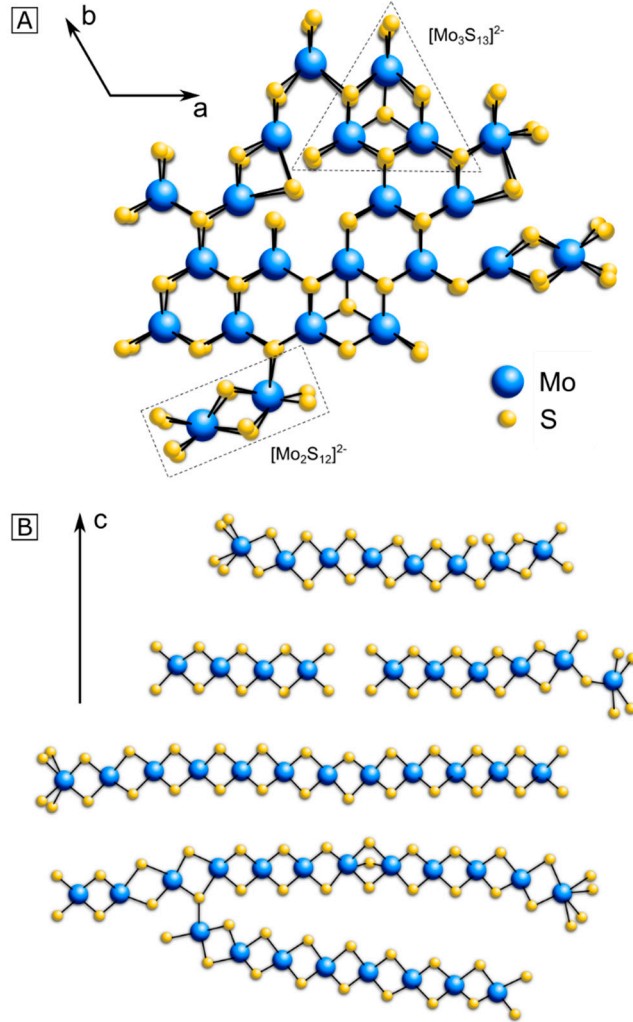

**Figure 7.** Schematic illustration of the suggested structure model for $MoS_{2+\delta}$. The indicated a-, b- and c-axis represent the unit cell and lattice parameters of $MoS_2$. (**A**): plane view; (**B**): out of plane view.

**Table 3.** Mo-Mo distance of $MoS_{2+\delta}$ compared with $MoS_2$ [30] and the cluster anions $[Mo_2S_{12}]^{2-}$ and $[Mo_3S_{13}]^{2-}$ [18].

| | $MoS_{2+\delta}$ | | | $MoS_2$ | $[Mo_2S_{12}]^{2-}$ | $[Mo_3S_{13}]^{2-}$ |
|---|---|---|---|---|---|---|
| | **Mean** | **Min** | **Max** | **[30]** | **[18]** | **[18]** |
| $d_{Mo\text{-}Mo}$ [Å] | $3.11 \pm 0.24$ | $2.83 \pm 0.21$ | $3.22 \pm 0.04$ | 3.16 | ≈2.8 | ≈2.7 |

Compared to the [100] direction, the variations in lattice spacings in the [001] direction are larger and the XRD analysis indicates an overall increased distance between the MoS2 layers, which are only bound to each other by weak Van der Waals interactions. The previously described in-plane variations and cluster-like disorder also can lead to local alternations in out of plane sulfur positions which might affect the Van der Waals bonding distance and thus induce a varying layer spacing. In particular, the alternation of the layer spacing at the nanocrystallite edges, as well as at the disclinations, is very large. In addition, external stress from boundaries to other neighboring crystals can induce further lattice spacing modulations.

Our suggested model is in qualitative agreement with literature results for disordered $MoS_x$ with $2 < x < 4$. $MoS_{2+\delta}$ is generally highly disordered. The in-plane structure of the detected nanocrystals partially features a hexagonal symmetry like $MoS_2$. The local defect structures show similarities to $[Mo_2S_{12}]^{2-}$ and $[Mo_3S_{13}]^{2-}$ clusters, which correlates well with the increased sulfur content compared to $MoS_2$.

Hinnemann et al. [7] studied $MoS_2$ nanoparticles with approximately 4 nm in diameter and 1 nm in apparent height on graphite and stated that only the edges of $MoS_2$ are interesting in the context of HER, as the basal plane of $MoS_2$ is catalytically inactive. Our structure model for $MoS_{2+\delta}$ strongly features frayed edges similar to nanocrystalline $MoS_2$ and due to the high defect concentration, coordinatively modified Mo sites also appear within the basal planes. Consequently, we expect that some of the active sites are similar to the report of Hinnemann et al. [7]. But in addition, cluster-like structures appear within the lattice planes as well as at their edges and in the disclinations. These planar structures exhibit a partial stacking and ordering along the c-axis, also establishing a similarity to the $MoS_2$ crystal, however, with increased lattice parameters c due to small crystal sizes as well as disorder in the in-plane structure. Typically, XPS for $MoS_x$ with $2 < x < 4$ indicates bridging and terminal disulfides as well as unsaturated molybdenum and sulfur ions [4,11,12,14,15]. The structure observed here featuring disordered nanocrystals with a size of a few nanometers can thus explain a high density of catalytically active sites which are present at the defective nanocrystal planes as well as at their edges. This, in consequence, could very well explain the much higher HER activity of $MoS_{2+\delta}$ compared to $MoS_2$.

Wu et al. [38] also report a reduced Mo–Mo distance of 2.778 Å for their amorphous $MoS_x$, which is in good agreement with this work (smallest Mo–Mo distance $(2.81 \pm 0.2)$ Å) and suggest it as a key feature for the higher activity as the electronic structure gets even more similar to the clusters. In addition, no influence of the sulfur dimer content on the activity was observed by Wu et al. [38]. This is in good agreement with this work, where the catalytic activity tends to correlate with Mo content. Considering also the high catalytic activity of disordered nanocrystalline $MoS_x$ with $x < 2$ reported by Xi et al. [37] this might indicate the larger impact of the presence of coordinately modified Mo sites and reduced lattice parameter rather than the S:Mo ratio. In addition, Ying et al. [29] reports on improved catalytic activity by increasing the concentration of sulfur vacancies. This supports to allocate the active site to the Mo edges. In addition, the structure model for nanocrystalline, partially disordered $MoS_{2+\delta}$ is based on stoichiometry compensating defects that change the Mo coordination and Mo–Mo bonding distance. Thus, both the S:Mo ratio and the processing induced microstructure influence the crystal structure.

## 4. Materials and Methods

All chemicals were purchased commercially and, if not stated otherwise, used without further purification. Deionized water (R = 18.2 MΩ) from an Elga Veolia PURELAB flex 4 water purification system was used for all experiments. Crystalline $MoS_2$ was purchased from Sigma-Aldrich.

### 4.1. Synthesis of $MoS_x$

Ammonium tetrathiomolybdate $(NH_4)_2[MoS_4]$ was prepared following a literature procedure by McDonald et al. [39]. The very high purity of the $(NH_4)_2[MoS_4]$ was confirmed by mass and Raman spectroscopy as well as XRD. To obtain amorphous molybdenum sulfide, a slightly modified method from the route of Li et al. [40] was used: $(NH_4)_2[MoS_4]$ (100 mg) was dissolved in water and hydrazine $(N_2H_4 \cdot H_2O, 0.3$ mL) was added. The mixture was transferred into a steal autoclave with a Teflon inlet (45 mL), heated up to 200 °C for 12 h and then allowed to cool down to room temperature. The reaction mixture was centrifuged (10 min @ 5000 rpm), washed with THF ($3 \times 25$ mL) as well as water ($2 \times 30$ mL) and freeze-dried. As comparably high amounts of $MoS_x$ were required for the XRD and TEM analyses, two batches of the product ($MoS_{2.6}$ and $MoS_{3.4}$) were synthesized due to the rather small volume of the available steel autoclave. Both $MoS_x$ batches were obtained as black powders in very similar yields of ~60 mg each.

### 4.2. Stoichiometry Determination

An Analytik Jena novAA® 350 flame atomic absorption spectrometer (F-AAS) (Analytik Jena, Jena, Germany) was used to determine S:Mo ratios. The calibration was performed using an ammonium heptamolybdate solution diluted by 0.03% v/v aqua regia. Prior to analysis, the samples (5 mg) were completely dissolved in aqua regia (5 mL) to oxidize molybdenum to Mo(VI). This solution was diluted to 200 mL and used without further dilution. The measurements were conducted five times for each sample and the mean value taken. The S:Mo ratios $2.6 \pm 0.13$ and $3.4 \pm 0.17$ were calculated from the initial sample weight and the F-AAS results presuming the absence of any other elements than molybdenum and sulfur. Another reason for the divergent composition could lie in the synthesis process. The addition of the reductant hydrazine is not performed in a controlled manner; neither is the solution stirred during the synthesis. This and the fact that the reaction mixture possibly contains high amounts of different sulfur species like sulfides, disulfides as well as hydrazine, ammonia and water in variable proportions could explain the differing S:Mo ratios.

X-ray fluorescence (XRF) (M4 Tornado from Bruker Corporation, Billerica, Massachusetts, USA) measurements calibrated with $MoS_2$ resulted in a similar S:Mo ratio of about 3. However, due to a strong peak overlap between molybdenum and sulfur, the resulting S:Mo ratio uncertainties are quite high when using XRF, energy-dispersive X-ray spectroscopy (EDS) (INCA detector from Oxford Instruments, Abingdon, England) or wavelength-dispersive X-ray spectroscopy (WDS) (JXA-8900 RL from JEOL, Akishima, Tokyo, Japan), which makes AAS the most reliable method in this case.

### 4.3. Electrochemistry

For the electrochemical measurements, a PRINCETON Applied Research Versa Stat 4 potentiostat (AMETEK Princeton Applied Research, Oak Ridge, TN, USA) was used. All measurements were performed in a three-electrodes setup using $MoS_2$ resp. $MoS_x$ on graphite sheets as working electrodes (WE), platinum as counter electrode (CE) in a separate compartment with a glass frit and an Ag/AgCl electrode as counter electrode (3 M KCl, RE). Sulfuric acid (0.5 M, pH 0.3) served as electrolyte. Cyclic voltammograms (CV) were recorded in a range of $0.1(-0.5)$ $V_{RHE}$ with a sweep rate of 20 mV s$^{-1}$. Tafel slopes were determined from the sixth cycle (first half) of the CV. All electrochemical measurements were *iR*-corrected at 85%.

### 4.4. Electrode Preparation

To eliminate impurities from the graphite sheet, the blank electrodes were cleaned with water, isopropanol and ethanol. The electrodes were prepared following procedures published by Cui et al. [41] 3 mg of the catalyst and 6 μL of Nafion solution (5 wt.%) were dispersed in 0.6 mL water/ethanol (49:50, v:v) and treated with ultrasound for 30 min. 48 μL of the ink was dropcasted onto an area of 1 cm × 1 cm of graphite sheet and dried under air at 60 °C for 1 h (catalyst loading $\sim 0.24 \frac{mg}{cm^2}$).

### 4.5. XRD Analysis

For the powder diffraction measurements, a Philips PW 1720 (Philips Analytical Technology GmbH) with a copper anode (λ = 1.541 Å) and a 5 × 0.1 mm line focus was used. The sample was placed in a pan made of brass. For data collection frames were measured for the duration of 10 s per step in 0.02° intervals of 2θ among 5° and 91°. The data were collected with by using a graphite secondary monochromator and a proportional counter.

### 4.6. TEM Lamella Preparation and TEM Analysis

For the TEM analysis 5 mg of powder ($MoS_{2+\delta}/MoS_2$) were dissolved in 1 mL of THF and then a 30 min long ultra-sonic treatment applied. One drop of this solution was then put on a carbon TEM Grid. The electron diffraction TEM investigations were performed with a Phillips CM12 (Philips Electron Optics GmbH) at 120 kV. For the high resolution TEM analysis a FEI Titan (FEI, Hillsboro, Oregon, USA) aberration corrected electron microscope with 300 keV was employed. For this microscope the information limit in high vacuum is about 0.08 nm.

### 4.7. Accuracy of Lattice Spacing Measurement in TEM

The TEM was calibrated using a gold reference sample to get a reliable accuracy. The dependence of the precision Δd/d on defocus induced changes of diffuseness of the FFT reflections is much smaller than the effect of a small diffraction vector length [42]. For each magnification, area of the FFT and measured d value, the precision Δd/d is calculated as a function of diffraction vector length and presented in Table S5. For the average values in Table 2, the weighted average of each lattice spacing $d_{hkl}$ is calculated from individual measurements, taking into account the error of the individual values. The total error is the statistical deviation from the weighted average plus the highest systematic error of the single measurements.

The TEM calibration was verified by using $MoS_2$ nanoparticles. The resulting values for $MoS_2$ are $d_{(100)}$ = (2.80 ± 0.06) Å and $d_{(001)}$ = (12.8 ± 0.7) Å. They are matching our XRD data and deviate from the literature values of $d_{(100)}$ = 2.737 Å and $d_{(001)}$ = 12.294 Å obtained by x-ray diffraction in Wildervanck et al. [30] by $\delta_{(100)}$ = 2.3% and $\delta_{(001)}$ = 4.2%. Within error, the determined $MoS_2$ lattice parameters are in agreement to these XRD results. Based on the calibration, the measurement of the locally reduced $d_{(100)}$ value of $MoS_{2+\delta}$ in this work is significant.

## 5. Summary

$MoS_{2+\delta}$ was prepared by solvothermal synthesis, immobilized on electrodes and electrochemically analyzed by cyclic voltammetry and chronopotentiometry. A comparison with $MoS_2$ confirmed the much higher activity of $MoS_{2+\delta}$. Hence the analyzed samples show the typical characteristics of $MoS_{2+\delta}$ which are reported in literature [4,11,14,18].

However, by means of HRTEM, XRD and electron diffraction, the structure of $MoS_{2+\delta}$ is assigned to a highly disordered variant of $MoS_2$ with very small nanocrystal size and cluster like local defects. The in-plane lattice parameter shows strong local variations and is reduced locally, whereas c-axis is increased in average. Due to the nanocrystalline structure, a high concentration of edges with changed Mo coordination and Mo–Mo distance is present. Furthermore, our HRTEM observations suggest that

disorder within the $MoS_2$ planes represent $[Mo_3S_{13}]^{2-}$ cluster like defects. Such structural features are also involved in the formation of disclinations. Altogether, this increases the ratio of coordinatively modified Mo and is in accordance to the scaling of electrochemical activity with Mo. Our results imply that S:Mo ratios > 2 are mainly important for the HER activity due to the processing induced nano- and defect structure.

**Supplementary Materials:** The following are available online at http://www.mdpi.com/2073-4344/10/8/856/s1, Figure S1: Rietveld refinement of $MoS_2$, Figure S2: Electron diffraction pattern of $MoS_{2.6}$ and $MoS_{3.4}$, Figure S3: HRTEM images of $MoS_2+\delta$ used for the lattice parameter analysis in Table S5, Figure S4: HRTEM images of $MoS_2$ and their FFTs, Figure S5: Reduced FFT of HRTEM image from Figure 5 A & B, Figure S6: SEM images of $MoS_2$ and $MoS_{3.4}$ powder, Figure S7: Cyclic voltammetry of the two $MoS_x$ samples and $MoS_2$, Table S1: Comparison of $d_{hkl}$ from XRD and SAD from Figures 2 and 3 for $MoS_{2+\delta}$ with $MoS_2$, Table S2: Goodness parameters and correction factor for the texture, Table S3: Refined lattice parameters and atom positions of $MoS_2$, Table S4: Overview of the most intense lattice planes in electron diffraction of $MoS_2$, Table S5: Result from the FFT analysis of HRTEM images of $MoS_{2+\delta}$.

**Author Contributions:** Conceptualization, E.R., S.H., P.K. and C.J.; investigation, E.R., M.-L.G., H.K., P.K., C.J.; data curation, E.R., S.H., M.-L.G.; writing—original draft preparation, E.R., S.H., M.-L.G. and C.J.; writing—review and editing, E.R., M.-L.G., P.K., C.J.; supervision, H.K., P.K. and C.J. All authors have read and agreed to the published version of the manuscript.

**Funding:** This research was funded by Deutsche Forschungsgemeinschaft (DFG), within the national priority program SPP 1613 "Fuels Produced Regeneratively Through Light-Driven Water Splitting".

**Acknowledgments:** The authors would like to thank Max Baumung for his contribution to the TEM sample preparation and Vladimir Roddatis for practical advices in TEM work.

**Conflicts of Interest:** The authors declare no conflict of interest.

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
