# Peer review of "Structure of Nanocrystalline, Partially Disordered MoS2+δ Derived from HRTEM—An Abundant Material for Efficient HER Catalysis"

_catalysts, doi:10.3390/catal10080856_

Round 1

Reviewer 1 Report

Authors present a very solid structural investigation on sulfur-rich MoSx particles synthesized by solvothermal method which are subsequently used as electrocatalysts towards hydrogen evolution reaction (HER) in sulfuric acid. Authors demonstrate that the MoSx (x › 2) derive from the crystalline, stoichiometric MoS2. In particular, these sulfur-rich MoSx particles show a partially disordered nanostructure with lower Mo-Mo distances. The occurrence of disclination defects is also proven. The paper is well-written and the findings are scientifically sound. I recommend publication of the manuscript after minor amendments, as indicated below:

1. I am not very much convinced about the particle analysis presented in the supplementary information file (Figure S6 and related text). I do not think that the shape of the MoS2+δ particles can be approximated to a sphere. Authors conclude that "Thus, the surface of the MoS2+δ electrode is by factor of ≈1.9 larger than the MoS2 surface due to particle size and shape." Well, I think this is a very rough estimate. Later, in the main manuscript, authors state that "Assuming the same degree of porosity, this indicates that the higher activity of the MoS2+δ electrodes compared to the MoS2 electrode cannot be explained by morphology effects alone." Is the degree of porosity of the resulting electrodes really the same? I mean, once the active material is placed on graphene, can be assure that the porosity is the same? In this case, determining the ECSA values would be preferred. 

2. In relation to Figure 1A, please indicate in the Experimental section the potential value from which the scanning is started. Is it 100 V vs. RHE? If so, I have a problem with the CV curves for the MoS2+δ based electrodes since either an anodic or a cathodic current is detected before 0 V vs RHE. Why is the origin of this current? Please, add arrows in Figure 1A to indicate the initial scan direction. I also recommend to include the response of the blank (MoSx-free graphene electrode).

3. "The long-term stability of the electrodes was tested in CP measurements (Figure 1 C)." It is Figure 1 B.

4. Typos: "The presents of only one sharp MoS2 reflex (002) in MoS2.6 [...]" (presence). "Bot samples (MoS2.6 & MoS3.4) show similar reduction and local variations in the Mo-Mo distance." (both)

Author Response

See word file

Reviewer 2 Report

There are major flaws in this manuscript and shouldn't be published. 

  1. Even with calibrated TEM, 5% error in measurement could not be avoided. The author used hrtem to precisely quantify the interplanar spacing, this could not be used in peer reviewed journal.
  2. In Figure 4, FFT too blurring, and any interplanar spacing from this FFT is not convincing. If the author wants to quantify the lattice spacing (within 5% error) by FFT, could do a filter on FFT and do the inverse FFT to show clear lattice spacing. But of course, it is still subject to 5% error. Any measurements based on this blurred FFT is not precise and misleading.
  3. Line 246-247, hrtem is not the proper technique to judge whether the material is order or disorder. Better option would be haadf stem imaging and imaging simulation. TEM image is composed of columns of atoms, and it is impossible to tell whether the materials is odered or disordered a TEM image. The judgement of order, disorder structure in this manuscript based on HRTEM is all misleading.
  4. Figure 6. Inaccurate measurements of lattice planes. Images in Figure 6 are severely defocused, any measurements based on Figure 6A is not accurate.
  5. line 213, author indicated lattice spacing to be 2.737A, there is no one TEM in the world could achieve resolution to be 0.0001A level, please be precise. 

Author Response

See word file

Reviewer 3 Report

The work by Christian et al. reports on the structure of nanocrystalline, partially disordered MoS2+x derived from HRTEM-an abundant material for efficient HER catalysts. Several effects are studied including sulfur, particle sizes, crystallinity, etc using SEM, XRD, HRTEM, and XRF. The use of MoS2+x for hydrogen evolution catalyst has been reported in the literature (ref. 4 and a few lists below) and a detailed investigation was performed in characterizing the effect of disordered nature of MoS2+x. The novelty of this work is very low for publication in current form

https://pubs.acs.org/doi/10.1021/jacs.6b03714

https://pubs.acs.org/doi/full/10.1021/acsaem.8b00973

1) Even though the author synthesized MoS2.6 and MoS3.4? Both synthesized materials through one method but they are not reproducible. Why didn’t the authors expand the work by making more materials through methods? This is one of the main concerns regarding the paper.

2) Move the experimental and characterization section after the introduction section.

Ammonium tetrathiomolybdate (NH4)2[MoS4] was prepared from the literature 37? Use the commercial (NH4)2[MoS4] to verify if you could synthesize similar MoS2 derivatives.

3) Role of Hydrazine in forming MoS2+x

4) The mole fractions of S and Mo within the materials should be verified with sophisticated techniques such as neutron activation analysis and their bending spectrum using XPS.

5) Sometimes TGA is also used to see the difference in mass loss of MoS2+X samples

6) Results sections and figures are randomly placed. They need to arrange in proper order.

7) This is 104  in agreement with the literature [4,6], where it is also described that disordered MoSx are far more 105  active HER catalysts than crystalline MoS2

Includes these values from the ref [4, 6] in Table SI along with others in the literature.

8) Concluding from 113  these results, a high sulfur content seems to be much less important for the HER catalysis rate than 114  the existence of a disordered structure.

https://pubs.acs.org/doi/10.1021/acsaem.8b00973

This MoS6 catalyst showed better HER behavior than MoS4. This statement is completely contradictory.

9) However, the different particle sizes and shapes result in an increase of the MoS2+δ surface by a factor 120  of only about 1.9 compared to MoS2 (see ESI for details).

In figure S6, the MoS2 sample showed varied shapes and sizes. Whereas MoS3.4 has nanosized particles along with aggregated micro size particles.  And SEM image of MoS2.6 is missing here

The author has considered the shape of MoS3.4 materials as spherical even though they have varied sizes and shapes, similar to MoS2. The inset in Figure S6B has poor resolutions.

What is the thickness of the MoS2 and as-synthesized MoS2+x? Please measure Raman and AFM to confirm the thickness

10) All of the measured values for overpotentials and Tafel slopes of the MoSx studied here cannot 131

 compete with the best MoSx that are known today (Table 1).

Extensive research has performed on MoSx based materials and the author only reported two refs. in Table 1. Please give credit to the earlier researchers.

11) The long-term stability of the electrodes was tested in CP measurements (Figure 1 C).

Replace Figure 1C to Figure 1B

12) For a set current density of 3 mA cm2, at the end of the experiments, MoS2.6 and MoS3.4 showed 143  overpotentials of ~210 mV and ~195 mV, respectively, in comparison to ~430 mV for MoS2

The overpotential of MoS2.6 and MoS3.4 is 250 and 255 mV but the Tafel slopes are reversed.

13) Authors need to calculate the total number of active sites and edges for the as-synthesized materials and MoS2

14) The XRD spectrum of as-synthesized materials are amorphous and diffraction peaks have broad patterns in Figure 2. How could the author finds the individual peaks from a broad pattern and comparing with the literature values shown in Table S1?

And the list goes on.

Author Response

See word file

Round 2

Reviewer 2 Report

The author answers all questions in detail, and I think it is convincing to publish now. 

Reviewer 3 Report

Accept